## [Decision Letter · Decision Letter 0]

14 Nov 2022

PONE-D-22-27487Tuberculosis treatment outcomes in Zimbabwe, 2021: The need to strengthen patient follow upPLOS ONE

Dear Dr. Chadambuka, 

Thank you for submitting your manuscript to PLOS ONE. After careful consideration, we feel that it has merit but does not fully meet PLOS ONE’s publication criteria as it currently stands. Therefore, we invite you to submit a revised version of the manuscript that addresses the points raised during the review process.

We look forward to receiving your revised manuscript.

Kind regards,

Limakatso Lebina, MBChB, Ph.D.

Academic Editor

PLOS ONE

Journal Requirements:

4. Please ensure that you refer to Figure 8 in your text as, if accepted, production will need this reference to link the reader to the figure.

5. Please upload a new copy of Figure 5 as the detail is not clear. Please follow the link for more information:

https://blogs.plos.org/plos/2019/06/looking-good-tips-for-creating-your-plos-figures-graphics/

https://blogs.plos.org/plos/2019/06/looking-good-tips-for-creating-your-plos-figures-graphics/

6. We note that Figure 5 in your submission contain map images which may be copyrighted. All PLOS content is published under the Creative Commons Attribution License (CC BY 4.0), which means that the manuscript, images, and Supporting Information files will be freely available online, and any third party is permitted to access, download, copy, distribute, and use these materials in any way, even commercially, with proper attribution. For these reasons, we cannot publish previously copyrighted maps or satellite images created using proprietary data, such as Google software (Google Maps, Street View, and Earth). For more information, see our copyright guidelines: http://journals.plos.org/plosone/s/licenses-and-copyright.

(1) You may seek permission from the original copyright holder of Figure 5 to publish the content specifically under the CC BY 4.0 license.  

**Additional Editor Comments:**

Thank you for sharing this interesting manuscript.

Please clarify or add the following:

Abstract

Methods - Statistical analysis is not included.

Results - Presumptive TB - on what basis? It would be important to include description in the abstract.

Main manuscript

Line 67: Explain what you mean by unfavorable outcomes due to not being evaluated.

In general, avoid the use of TB cases, and rather refer to People with TB - recommended appropriate phrases according to the words matter by the STOP TB PARTNERSHIP.

Methods: Lines 85-85 - provide proportion of cure provided by the public sector.

Results: data Quality, need to explain why more cases were evaluated compared to reported.

TB negative cases are usually not included in TB care cascades.

Explain the basis for 99.1% treatment initiation rate.

Reviewers' comments:

Reviewer's Responses to Questions

**Comments to the Author**

1. Is the manuscript technically sound, and do the data support the conclusions?

Reviewer #1: Partly

Reviewer #2: Yes

2. Has the statistical analysis been performed appropriately and rigorously? 

Reviewer #1: Yes

Reviewer #2: Yes

3. Have the authors made all data underlying the findings in their manuscript fully available?

The PLOS Data policy requires authors to make all data underlying the findings described in their manuscript fully available without restriction, with rare exception (please refer to the Data Availability Statement in the manuscript PDF file). The data should be provided as part of the manuscript or its supporting information or deposited to a public repository. For example, in addition to summary statistics, the data points behind means, medians and variance measures should be available. If there are restrictions on publicly sharing data—e.g., participant privacy or use of data from a third party—those must be specified.

Reviewer #1: Yes

Reviewer #2: No

4. Is the manuscript presented in an intelligible fashion and written in standard English?

Reviewer #1: Yes

Reviewer #2: Yes

5. Review Comments to the Author

Reviewer #1: Thank you for the opportunity to review the manuscript “Tuberculosis treatment outcomes in Zimbabwe, 2021: The need to strengthen patient follow-up” by Christwish and colleagues. The manuscript presents interesting and informative data on the TB disease care cascade in Zimbabwe, as determined through a review of nationally-reported data. These data do highlight several areas that could be strengthened for care of patients with TB in Zimbabwe, including in data reporting and follow-up. I have several suggestions which will hopefully help to strengthen the manuscript.

Overall:

- The majority of the discussion section focuses on the geospatial link between provinces with mining communities and death/LTFU in the cohort, and the authors list this finding as a key take-away from their manuscript. The authors should consider the following points:

o Currently, the manuscript does not actually present data on mining communities in Zimbabwe. To make their claim compelling, the authors should present data on the distribution of mining communities in Zimbabwe, such as the proportion of each province’s population engaged in ASM or other data that the authors may be aware of. Ideally, the authors would also present statistics supporting the association between ASM and adverse TB outcomes. Otherwise, the hypothesis that ASM is linked to adverse TB outcomes is speculative in this paper—it may be a hypothesis worthy of future study (and, as the authors note, it justified by other prior research), but is not actually evaluated in this manuscript.

o The claim in the discussion that ASM is linked to adverse TB outcomes because adverse TB outcomes occur more frequently in provinces with high ASM is subject to an ecological fallacy. Because not all individuals living in provinces with high rates of ASM are miners, it is not necessarily correct to claim that ASM causes adverse TB outcomes without doing additional research to support this claim (e.g., directly measuring these outcomes among miners, while controlling for other factors).

- Typos and unusual syntax occasionally interfered with interpretation of the manuscript, and a careful edit would enhance readability.

- The authors should harmonize their use of significant digits when reporting proportions and percentages.

Title

- The title references “2021”; however, the data reported are from 2019 and 2020. I suggest that the authors either drop the year from the title or change it to better reflect the data presented.

Abstract

- Introduction:

o Line 27 - The first sentence of the abstract introduction seems to present the primary finding of this paper. Is this information already known (in which case, it would be helpful to describe how this paper builds on this information)?

- Methods:

o Line 31 - It would be helpful to define DHIS2.3 here.

- Results:

o Line 40-41 - The finding that “only four of 11 provinces were above the national average success rate” simply indicates that these data are skewed: by definition there will always be areas above and below the average; fewer areas above the average only indicates increased skewing of the data. I’m not sure how meaningful this finding is unto itself, unless there is a success rate that is benchmarked to the average that Zimbabwe aims for.

o Line 43 – see discussion of ASM above.

Introduction

- Line 53 – I think the authors mean to state that “1.5 million people died from TB and 10 million people developed TB in 2020” (per the WHO 2021 World TB Report) (as stated now, it sounds like 15% of the world’s population died from TB in 2020).

- Line 55 – the authors should make clear that 85-89% of “TB globally” occurred in the top 30 TB countries, not in Zimbabwe alone.

- Line 57-65 – this paragraph presents methods for TB treatment cohort review. I believe it would fit more appropriately in the methods section

- Line 72 – who receives training?

- Line 76 – I think the authors could further justify why they investigated the spatial distribution of treatment outcomes. Did they have hypotheses about different spatial distributions? Did this analysis serve an administrative or policy purpose, which is why it was pursued?

Materials and methods

- How were “presumptive” TB cases identified?

- A variety of different numerators and denominators are used in the analysis, including “deaths of all forms of TB” (line 114) and “new pulmonary bacteriologically confirmed TB cases” (line 117), but it seems as if these disparate groups were added together to attain the overall proportions described in the cascade. Notably, the authors also state that they excluded patients with extrapulmonary TB, but then what is meant by “deaths of all forms of TB”? I am left wondering if these different groups should not have been added together to describe the overall cascade, or if the description of the outcome proportions and exclusions could be further clarified?

- Line 80 – the authors discuss characteristics of the administrative province, but the geospatial analysis in the results presents data at the “province district” level. It may be helpful to add a description of the districts as well, or depict only provinces in the maps.

- Line 97-98 – why did the authors choose 2019 to study DR-TB and 2020 to study DS-TB? Were there changes due to COVID-19 that affected reporting of TB cases or function of Zimbabwe’s TB control program, which may affect our interpretation of the DS-TB results (particularly in relation to the pre-COVID DR-TB results)?

- Line 100, 105, 106 – it may be more accurate to state “study outcomes” instead of “study variables”

- Line 114, 116, 119, 122 – the authors state that they are computing “proportions” (which range from 0-1), but are actually reporting “percentages” (which range from 0-100%). This should be corrected here and in the figures.

- Line 139-141 – Why did the authors elect to present maps at the district level, while discussing other results at the provincial level?

Results

- Consider including an overall demographics table describing the characteristics of included patients

- Line 149 – I am confused as to why there was a discrepancy in the number of patients with outcomes evaluated and cases notified. Reviewing Table 1, it appears that in some provinces, more outcomes were evaluated than cases notified, and in other provinces, the pattern was reversed. My confusion probably arises from my lack of understanding of how “evaluated” and “notification” differ, but I suspect that other readers may be similarly confused. I think it would be helpful to provide additional explanation.

- Line 149/Table 1 – when I add the rows in the “outcomes evaluated” column of Table 1, I get 16115, instead of 16127 (as stated in the text).

- Line 155 – it is not correct to say that pre-diagnosis LTFU was 31.4% (achieved by adding 10% to 21.4%), because these two steps have different denominators. I believe the pre-diagnosis LTFU rate that the authors should present is (107583-76066)/(107583) = 29.3%.

- Line 163 – “boarder” should be “border”. It is also not clear to me from the maps that high death/LFTU rates were higher in border districts than in other districts in Zimbabwe.

- Line 165 – why was the DR-TB target 740 cases?

- Line 166, 169, 174 – these figure numbers are all off by 1.

- Line 175-177 – the authors present results of an inquiry into results, but the methodology for this inquiry is not presented. Were these interviews? Surveys? What questions were asked and what were the hypotheses? Also, it would be helpful to be explicit about what “possible reasons” means. Also, what “other outcomes” (line 176) were evaluated, and why?

- Line 178-179 – this sentence seems speculative, and probably belongs in the discussion because no data were measured in this study that directly support this statement.

Discussion

- Line 183-184 – there are no data that are actually presented showing high death rates in provinces with ASM (without readers knowing a priori which provinces have high rates of ASM), and it is not clear why the authors focus on ASM in these provinces and not other factors that may be relevant (e.g. HIV prevalence, number/access to health centers, administrative factors affecting reporting of TB data, or others). Therefore, I don’t think the authors should list this as a key finding, unless additional analysis is performed (per above).

- Line 185 (“some reasons given”) – are these the authors’ hypotheses about why non-evaluation rates were high? Or are these the results of inquiries into the reporting system?

- Line 194-209 – see above regarding concerns about linking ASM with TB outcomes in this study (even if this hypothesis is supported in other studies).

Limitations

- I think it would be helpful for the authors to list and discuss other potential limitations in this study. These include:

o Challenges with data quality (mentioned elsewhere in the manuscript, but not in the limitations section)

o Lack of information about re-infection (which is known to be a major gap in the TB disease care cascade (see for example: https://pubmed.ncbi.nlm.nih.gov/32072022/)

o Misattribution bias – could patients in the “not evaluated” category have been evaluated but not reported (e.g. at a different health center)? Thus is there a possibility that use of this outcome measure bias the study’s findings?

o Discordant timing of the study – could inclusion of patients from years before and after COVID-19 onset affect these results or interpretation of the key findings?

Conclusions

- The conclusions seem to focus on data quality, which is not extensively discussed in the results or discussion section (there is only 1 sentence in the discussion about data quality). If an appeal for improved data quality is the authors’ key point, then I think it should be further described in the rest of the manuscript, and particularly in the discussion section.

Tables/figures

- Figure 3, Figure 4, Figure 5, Figure 7, Figure 8 – I believe these should list “percentage of cases” and not “proportion of cases”

- Figure 4, Figure 8 – the percentages listed atop each of the bars often overlap with adjacent bars and numbers, making them difficult to read. Consider removing these caps or spacing the bars more widely to improve legibility.

Reviewer #2: Tuberculosis treatment outcomes in Zimbabwe, 2021: The need to strengthen patient follow up

This is a descriptive analysis of routine nationwide tuberculosis data from Zimbabwe. From the manuscript and presented resulted, it can be inferred that the authors analyzed aggregate data rather than case based data. However, this important information is not explicitly mentioned in the methods. Probably the authors assume that all readers are familiar with the format of the district health information system (DHIS) from which their data was obtained. Using aggregate data is an important limitation of this study because characteristics of patients are not available for analysis. If case based data was used, one would have expected detailed analyses of patient characteristics and predictors of unfavorable outcomes. Nevertheless, I believe that health facility characteristics some of which are described in the methods (ownership, level/tier etc) could have enriched the analysis. Secondly, this analysis included two separate cohorts from two time periods complicating the interpretation of findings. The presented results are also not very clear on the time periods – Table 1 does not mention the time period for the notifications and whether the notifications were drug resistant or sensitive; Figure 2 only mentions the year 2020. Perhaps the authors could focus on drug sensitive TB and conduct a more thorough analysis. Additionally, the following issues should be addressed:

- The types of TB (extrapulmonary, pulmonary etc) are not presented. Some types of TB are treated for longer periods than others. Depending on when this analysis was conducted, for patients who initiated TB treatment in late 2020 and were to be treated for longer than 6 months may not have completed treatment. These could have been categorized as “not evaluated”. How did the authors ensure that only patients who should have completed treatment at the time of analysis were excluded?

- The description of the reporting system for TB data should clearly state that only aggregate data is transmitted based on certain pre-defined indicators. The list of indicators that are routinely reported should be mentioned.

- Only a list of TB treatment outcome variables are described yet analyses include presumptive TB cases. The sources and variables for the presumptive analyses are not described.

- There is a mix up in the methods section. The section entitled “study variables” presents a list of how analyses were done. This section should be merged with the “data analysis” section and restructured to be a description rather than a list.

- The data analysis section attempts to describe the presumptive TB cascade rather than describing how analyses were conducted. The paragraph talking of this cascade is also confusing with a statement suggesting that smear microscopy should be excluded in the presumptive TB cascade followed by a statement stating that smear microscopy was included.

- Additional analyses could have been conducted to explore if some aggregate variables were associated with unfavorable outcomes. In the results and discussions the authors suggest that residence in certain regions was associated with unfavorable outcomes yet no formal testing of such analyses were performed.

- The results could have been presented in a sequential manner showing how many participant enter each step of the cascade and how many successfully transitioned to the next step. A flow diagram might have been very useful in illustrating this. The cascade could include steps such as presumptive TB, investigated, diagnosed, treatment initiated, treatment completion. This would assist the authors to better describe their results.

- The first paragraph of the discussion reads like a list rather than a high level summary of key findings.

- This study using routine data had profound limitations that should be discussed elaborately. Currently, the limitations section is too brief.

6. PLOS authors have the option to publish the peer review history of their article (what does this mean?). If published, this will include your full peer review and any attached files.

Reviewer #1: No

Reviewer #2: No

---

## [Author Response · Author response to Decision Letter 0]

11 Jan 2023

Dear Editor(s)

Thank you for reviewing our manuscript and for the constructive comments.

Please find the responses to comments raised in our submitted manuscript ‘Tuberculosis treatment outcomes in Zimbabwe: The need to strengthen patient follow up’

Comment number Editor’s comments Response 

Comment 1. Please ensure that your manuscript meets PLOS ONE's style requirements, including those for file naming. 

Response: We have followed the style requirements

Comment 2. Please provide additional details regarding participant consent. In the ethics statement in the Methods and online submission information, please ensure that you have specified what type you obtained (for instance, written or verbal, and if verbal, how it was documented and witnessed). If your study included minors, state whether you obtained consent from parents or guardians. If the need for consent was waived by the ethics committee, please include this information. 

Response: The study was reviewed and approved by the National TB/HIV Institutional Review Board. All identifiable data was deidentified for analysis to ensure confidentiality. We obtained written informed consent from key informants. We obtained permission to conduct the analysis from the Secretary for Health and Child Care, and the Health Studies Office (HSO). 

Comment 3. In your Data Availability statement, you have not specified where the minimal data set underlying the results described in your manuscript can be found. PLOS defines a study's minimal data set as the underlying data used to reach the conclusions drawn in the manuscript and any additional data required to replicate the reported study findings in their entirety. All PLOS journals require that the minimal data set be made fully available. 

Response: We have attached the minimal dataset

10.6084/m9.figshare.21737972

Comment 4. Please ensure that you refer to Figure 8 in your text as, if accepted, production will need this reference to link the reader to the figure. 

Response: We referenced Figure 8 in the text.

Comment 5. Please upload a new copy of Figure 5 as the detail is not clear. 

Response: We have worked on the map and made it clearer. The new version has been uploaded. 

Comment 6. We note that Figure 5 in your submission contain map images which may be copyrighted. All PLOS content is published under the Creative Commons Attribution License (CC BY 4.0), which means that the manuscript, images, and Supporting Information files will be freely available online, and any third party is permitted to access, download, copy, distribute, and use these materials in any way, even commercially, with proper attribution. For these reasons, we cannot publish previously copyrighted maps or satellite images created using proprietary data, such as Google software (Google Maps, Street View, and Earth). 

Response: We created the maps using ArcGIS software therefore there are no copyright challenges.

Abstract 

Methods 

Comment: Statistical analysis is not included. 

Response: We calculated means, medians, and frequencies 

Results 

Comment: Presumptive TB - on what basis? It would be important to include description in the abstract. 

Response: Patients were presumed to have TB based on symptom screening and or chest x-ray

Main manuscript 

Comment: Line 67 Explain what you mean by unfavorable outcomes due to not being evaluated. 

Response: We acknowledge that the statement is vague. The introduction now reads, “Zimbabwe has a high proportion of not evaluated TB treatment outcomes”

 Comment: In general, avoid the use of TB cases, and rather refer to People with TB - recommended appropriate phrases according to the words matter by the STOP TB PARTNERSHIP. 

Response: The phrase has been edited to People with TB throughout the manuscript.

Methods

Lines 85-85 

Comment: Provide proportion of cure provided by the public sector. 

Response: The public sector provides 95% of all health services. All data analyzed in this study were accessed from the public domain. All patients diagnosed with TB from the private sector are referred to the public sector for notification and treatment.

Results 

 Comment: Data Quality, need to explain why more cases were evaluated compared to reported. 

Response: Some reasons for the discrepancies included 1) facilities evaluated outcomes for patients who were transferred in while the same patients’ outcomes were evaluated at notifying facilities. 2) Not evaluating transfer out cases which in turn were not evaluated by the receiving facilities. 3) Other facilities evaluated outcomes and the findings are either entered in wrong/ outdated datasets or the outcome was not documented. The length of treatment for DR-TB could also contribute to loss to follow up and non-evaluation of outcomes as it may be difficult to follow up patients for long periods.

 TB negative cases are usually not included in TB care cascades. The study also intended to identify where patients are lost to follow up hence starting the cascade from presumption 

 Comment: Explain the basis for 99.1% treatment initiation rate. 

Response: Of the 9888 people who tested positive for TB, the program was able to initiate 9795 and this equates to a 99.1% treatment initiation rate. The target treatment initiation rate was 100%.

 Reviewer 1 comments 

Comment: The majority of the discussion section focuses on the geospatial link between provinces with mining communities and death/LTFU in the cohort, and the authors list this finding as a key take-away from their manuscript. The authors should consider the following points: Currently, the manuscript does not actually present data on mining communities in Zimbabwe. To make their claim compelling, the authors should present data on the distribution of mining communities in Zimbabwe, such as the proportion of each province’s population engaged in ASM or other data that the authors may be aware of. Ideally, the authors would also present statistics supporting the association between ASM and adverse TB outcomes. Otherwise, the hypothesis that ASM is linked to adverse TB outcomes is speculative in this paper—it may be a hypothesis worthy of future study (and, as the authors note, it justified by other prior research), but is not actually evaluated in this manuscript. 

Response: -According to the 2012 population census report: artisanal and small-scale miners are largely populated in Midlands (19457), Mashonaland West (13267), Matabeleland South (12153), and Mashonaland Central provinces. The same provinces have a high prevalence of TB in ASMs, high death rates, and loss to follow-up.

-Prevalence of TB in ASMs was 6.8% in Zimbabwe (Moyo, et al, 2022)

-ASMs are highly mobile, work in remote areas with poor access to health services which contributes to late diagnosis which is associated with poor prognosis, and high mobility contributes to high losses to follow-up.

 Comment: The claim in the discussion that ASM is linked to adverse TB outcomes because adverse TB outcomes occur more frequently in provinces with high ASM is subject to an ecological fallacy. Because not all individuals living in provinces with high rates of ASM are miners, it is not necessarily correct to claim that ASM causes adverse TB outcomes without doing additional research to support this claim (e.g., directly measuring these outcomes among miners, while controlling for other factors). 

Response: We agree that the claim is subject to ecological fallacy. However, the statement was an opinion, or hypothesis and a possible explanation for the findings. A recommendation for further research on the link has been included.

 Comment: Typos and unusual syntax occasionally interfered with interpretation of the manuscript, and a careful edit would enhance readability. 

Response: We have revised and edited the manuscript

 Comment: The authors should harmonize their use of significant digits when reporting proportions and percentages. 

Response: We have harmonized our proportions and percentages to one significant figure.

Comment: Title The title references “2021”; however, the data reported are from 2019 and 2020. I suggest that the authors either drop the year from the title or change it to better reflect the data presented. 

Response: We concur with your guidance and have dropped the year in the topic

Abstract 

Introduction 

Comment: Line 27 - The first sentence of the abstract introduction seems to present the primary finding of this paper. Is this information already known (in which case, it would be helpful to describe how this paper builds on this information)? 

Response: The introduction has been edited and now reads: ‘A preliminary review of TB treatment outcomes indicated that Mutare City had 22% of outcomes not evaluated in quarter 2 of 2021, and approximately 40% of 2021 outcomes for Harare uniformed forces were not evaluated. The problem persists despite training on data analysis and use. We analysed the TB cohorts to determine the presumptive cascade and the spatial distribution of treatment outcomes in Zimbabwe.’

Methods 

Comment: Line 31 - It would be helpful to define DHIS2.3 here. 

Response: We have defined DHIS 2.3. It means District Health Information Software version 2.3

Results 

Comment: Line 40-41 - The finding that “only four of 11 provinces were above the national average success rate” simply indicates that these data are skewed: by definition there will always be areas above and below the average; fewer areas above the average only indicates increased skewing of the data. I’m not sure how meaningful this finding is unto itself, unless there is a success rate that is benchmarked to the average that Zimbabwe aims for. Response: Thank you for the guidance. The statement has been reworded to “seven out of 11 provinces performed below the targeted success rate of 64%. 

 Comment: Line 43 – see discussion of ASM above The concerns raised have been addressed in above sections

Body Introduction 

Comment: Line 53 – I think the authors mean to state that “1.5 million people died from TB and 10 million people developed TB in 2020” (per the WHO 2021 World TB Report) (as stated now, it sounds like 15% of the world’s population died from TB in 2020). 

Response: We have revised the statement. The statement now reads “1.5 million people died from TB out of the 10 million people who developed TB in 2020” 

 Comment: Line 55 – the authors should make clear that 85-89% of “TB globally” occurred in the top 30 TB countries, not in Zimbabwe alone. 

Response: We have revised the statement. The statement now reads “Zimbabwe remains one of the top 8 countries in Africa on the world’s top 30 list of countries heavily burdened by TB/HIV and MDR-TB. The high-burden countries account for 85-89% of people with TB globally”

 Comment: Line 57-65 – this paragraph presents methods for TB treatment cohort review. I believe it would fit more appropriately in the methods section 

Response: The authors thought it would not fit in the methods section because we did not conduct a cohort review process but rather, we analysed the results of the cohort review process. 

 Comment: Line 72 – who receives training? 

Response: We have qualified the statement so that it is clear. It is healthcare workers involved in TB that receive training.

 Comment: Line 76 – I think the authors could further justify why they investigated the spatial distribution of treatment outcomes. Did they have hypotheses about different spatial distributions? Did this analysis serve an administrative or policy purpose, which is why it was pursued? 

Response: The authors investigated the spatial distribution of treatment outcomes to inform programming and aid in crafting targeted interventions to the problem.

-The analysis serves administrative and policy purposes to enable targeted interventions by administrative (geographic) area. 

Materials and methods 

Comment: How were “presumptive” TB cases identified? 

Response: Patients are presumed based on symptomatic screen and/or chest x-ray

 Comment: A variety of different numerators and denominators are used in the analysis, including “deaths of all forms of TB” (line 114) and “new pulmonary bacteriologically confirmed TB cases” (line 117), but it seems as if these disparate groups were added together to attain the overall proportions described in the cascade. Notably, the authors also state that they excluded patients with extrapulmonary TB, but then what is meant by “deaths of all forms of TB”? I am left wondering if these different groups should not have been added together to describe the overall cascade, or if the description of the outcome proportions and exclusions could be further clarified? 

Response: Thank you for the observation. the formula was misplaced, and it has been rectified. The correct formula we used included all forms of TB. -patients with extrapulmonary TB were initially excluded and later included in the final analysis. Leaving the statement as it is, was an oversight on our part.

 Comment: Line 80 – the authors discuss characteristics of the administrative province, but the geospatial analysis in the results presents data at the “province district” level. It may be helpful to add a description of the districts as well, or depict only provinces in the maps Response: Districts have been described in the manuscript. ‘These provinces are further divided into 91 administrative districts from the recent delimitation activities. However, there are still 73 health districts, which offer primary and secondary level care health services.’

 Comment: Line 97-98 – why did the authors choose 2019 to study DR-TB and 2020 to study DS-TB? Were there changes due to COVID-19 that affected reporting of TB cases or function of Zimbabwe’s TB control program, which may affect our interpretation of the DS-TB results (particularly in relation to the pre-COVID DR-TB results)? 

Response: Treatment outcomes are evaluated quarterly, and the cohort analyzed is those clients enrolled during the same period the previous year for drug-sensitive TB and the previous two years for drug-resistant TB (because some patients are still on long regimens which may take up to 24 months to complete treatment). The outcomes recorded in the 2021 dataset are outcomes for the 2019 DRTB and 2020 DSTB cohorts. COVID-19 affected the TB program significantly. However, the interpretation of the results may not be affected since the findings for the two cohorts were described/reported separately and were not compared in any way. 

 Comment: Line 100, 105, 106 – it may be more accurate to state “study outcomes” instead of “study variables” 

Response: We have renamed them to study outcomes

 Comment: Line 114, 116, 119, 122 – the authors state that they are computing “proportions” (which range from 0-1), but are actually reporting “percentages” (which range from 0-100%). This should be corrected here and in the figures. 

Response: The authors have rectified and ensured that all are expressed as percentages.

 Comment: Line 139-141 – Why did the authors elect to present maps at the district level, while discussing other results at the provincial level? Response: We intended to get a more specific picture of whether the undesirable outcomes were contributed by all districts or a few districts within the provinces were not performing well. 

Results 

Comment: Consider including an overall demographics table describing the characteristics of included patients 

Response: The demographics table has been added and it shows notifications by sex, age, and type of TB

 Comment: Line 149 – I am confused as to why there was a discrepancy in the number of patients with outcomes evaluated and cases notified. Reviewing Table 1, it appears that in some provinces, more outcomes were evaluated than cases notified, and in other provinces, the pattern was reversed. My confusion probably arises from my lack of understanding of how “evaluated” and “notification” differ, but I suspect that other readers may be similarly confused. I think it would be helpful to provide additional explanation Ideally, there should not be discrepancies between notified cases and evaluated outcomes.

Response: An inquiry of possible reasons for the inconsistencies was made through key informant interviews, and some reasons highlighted were --that facilities were not evaluating transfer-out cases which in turn will not be evaluated by the receiving facilities. 

- outcomes were evaluated, and the findings were entered in wrong/ outdated datasets

-Outcomes were evaluated but were not documented. 

-Some health workers reported to have forgotten to evaluate treatment outcomes due to workload

- 

Comment: Line 149 Table 1 – when I add the rows in the “outcomes evaluated” column of Table 1, I get 16115, instead of 16127 (as stated in the text). 

Response: Thanks for the correction. We realized that there was a typing and calculation error. Mashonaland west province evaluated 2184 outcomes not 2148 as indicated earlier. The total is now 16151. We rectified the typing and calculation errors. 

Comment: Line 155 – it is not correct to say that pre-diagnosis LTFU was 31.4% (achieved by adding 10% to 21.4%), because these two steps have different denominators. I believe the pre-diagnosis LTFU rate that the authors should present is (107583-76066)/ (107583) = 29.3%. 

Response: Thank you for the correction. We have rectified it.

 Comment: Line 163 – “boarder” should be “border”. It is also not clear to me from the maps that high death/LFTU rates were higher in border districts than in other districts in Zimbabwe. 

Response: The spelling has been corrected

-The darker shaded areas are more prevalent in areas that are in and around the border districts.

 Comment: Line 165 – why was the DR-TB target 740 cases? 

Response: -The target was set based on estimates from the last DR-TB prevalence survey (2016)

 Comment: Line 166, 169, 174 – these figure numbers are all off by 1. 

Response: We have rectified the figure numbering and now these sections have figure 6,7 and 8 respectively

 Comment: Line 175-177 – the authors present results of an inquiry into results, but the methodology for this inquiry is not presented. Were these interviews? Surveys? What questions were asked and what were the hypotheses? Also, it would be helpful to be explicit about what “possible reasons” means. Also, what “other outcomes” (line 176) were evaluated, and why? -We have included that we conducted key informant interviews to determine why they were discrepancies in the numbers notified and the outcomes evaluated.

Response: -Question asked was “may you please shed light on the discrepancies between number notified and outcomes evaluated?”

- Possible reasons simply meant reasons for discrepancies. We have reworded the statement in the manuscript

-No other outcomes were evaluated. However, the statement meant to indicate that other health facilities evaluated outcomes and entered the findings in wrong datasets or some facilities entered the wrong number notified alongside the outcomes evaluated.

 Comment: Line 178-179 – this sentence seems speculative, and probably belongs in the discussion because no data were measured in this study that directly support this statement. -Response: The authors have moved the statement to discussion section.

Comment: Discussion Line 183-184 – there are no data that are actually presented showing high death rates in provinces with ASM (without readers knowing a priori which provinces have high rates of ASM), and it is not clear why the authors focus on ASM in these provinces and not other factors that may be relevant (e.g. HIV prevalence, number/access to health centers, administrative factors affecting reporting of TB data, or others). Therefore, I don’t think the authors should list this as a key finding, unless additional analysis is performed (per above). -The authors have included data on ASM. Additional discussion points have been included on HIV prevalence, and access to health facilities. 

Response: According to the 2012 population census report: artisanal and small-scale miners are largely populated in Midlands (19457), Mashonaland West (13267), Matabeleland South (12153), and Mashonaland Central provinces. 

-There is a triple burden of TB, HIV, and silicosis in ASMs. Prevalence of TB in ASMs was 6.8%, HIV and silicosis 18.0% respectively in Zimbabwe (Moyo, et al, 2022)

-ASMs are highly mobile, work in remote areas with poor access to health services which contributes to late diagnosis which is associated with poor prognosis, and high mobility contributes to high losses to follow-up.

 Comment: Line 185 (“some reasons given”) – are these the authors’ hypotheses about why non-evaluation rates were high? Or are these the results of inquiries into the reporting system? 

Response: -These are results of enquiries into the reporting system (explanations to data discrepancies)

 Comment: Line 194-209 – see above regarding concerns about linking ASM with TB outcomes in this study (even if this hypothesis is supported in other studies). Response:

-We have included data on ASM in earlier responses

There are more ASMs in Midlands, Matabeleland South, Mashonaland West, and Mashonaland Central provinces. ASMs are mobile, have poor access to health services. contributes to late diagnosis which is associated with poor prognosis, and high mobility contributes to high losses to follow-up.

Limitations 

Comment: I think it would be helpful for the authors to list and discuss other potential limitations in this study. These include: 

-Challenges with data quality (mentioned elsewhere in the manuscript, but not in the limitations section)

-Lack of information about re-infection (which is known to be a major gap in the TB disease care cascade (see for example: https://pubmed.ncbi.nlm.nih.gov/32072022/)

-Misattribution bias – could patients in the “not evaluated” category have been evaluated but not reported (e.g. at a different health center)? Thus is there a possibility that use of this outcome measure bias the study’s findings?

- Discordant timing of the study – could inclusion of patients from years before and after COVID-19 onset affect these results or interpretation of the key findings? 

Responses: We have included challenges with data quality in limitations

-We have included lack of information about re-infection in limitations

-We have included misattribution bias in limitations

-Treatment outcomes are evaluated quarterly, and the cohort analyzed is those clients enrolled during the same period the previous year for drug-sensitive TB and the previous two years for drug-resistant TB (because some patients are still on long regimens which may take up to 24 months to complete treatment). The outcomes recorded in the 2021 dataset are outcomes for the 2019 DRTB and 2020 DSTB cohorts. COVID-19 affected the TB program significantly. However, the interpretation of the results may not be affected since the findings for the two cohorts were described/reported separately and were not compared in any way. 

Conclusion The conclusions seem to focus on data quality, which is not extensively discussed in the results or discussion section (there is only 1 sentence in the discussion about data quality). If an appeal for improved data quality is the authors’ key point, then I think it should be further described in the rest of the manuscript, and particularly in the discussion section. We included discussion on data quality in the manuscript

Tables and figures Figure 3, Figure 4, Figure 5, Figure 7, Figure 8 – I believe these should list “percentage of cases” and not “proportion of cases” Thank you. We have rectified and changed to percentage of cases

 Figure 4, Figure 8 – the percentages listed atop each of the bars often overlap with adjacent bars and numbers, making them difficult to read. Consider removing these caps or spacing the bars more widely to improve legibility. We have edited the graphs

 Reviewer 2 comments 

Overall This is a descriptive analysis of routine nationwide tuberculosis data from Zimbabwe. From the manuscript and presented resulted, it can be inferred that the authors analyzed aggregate data rather than case based data. However, this important information is not explicitly mentioned in the methods. Probably the authors assume that all readers are familiar with the format of the district health information system (DHIS) from which their data was obtained. Using aggregate data is an important limitation of this study because characteristics of patients are not available for analysis. If case based data was used, one would have expected detailed analyses of patient characteristics and predictors of unfavorable outcomes. Nevertheless, I believe that health facility characteristics some of which are described in the methods (ownership, level/tier etc) could have enriched the analysis. -Thank you for the feedback we have included in the methods the following statement, ‘This is a descriptive analysis of routine nationwide aggregate tuberculosis data from Zimbabwe.

-We also noted and included the limitation that aggregate data was used 

- We chose to present data by province because the province is the basic management unit in the TB program. 

-Analysis by ownership may not apply to our setting because all health facilities are registered under the ministry of health and childcare. Therefore, the government assumes ownership of all health facilities. 

-Analysis by tier will be good. However, recommendations will not be targeted since the basic unit of program management are provinces in our setting. We propose an analysis of the provincial data by rural or urban status since management systems in these settings differ.

 Secondly, this analysis included two separate cohorts from two time periods complicating the interpretation of findings. The presented results are also not very clear on the time periods – Table 1 does not mention the time period for the notifications and whether the notifications were drug resistant or sensitive; Figure 2 only mentions the year 2020. Perhaps the authors could focus on drug sensitive TB and conduct a more thorough analysis. -We included the period for notifications in Table 1 and indicated that it was drug-sensitive TB

-We also included that it is drug-sensitive TB in Figure 2

-We concur with the given suggestion and have focused our analysis on DSTB because we realise the presence of both may introduce difficulties in understanding the message.

Additionally, the following issues should be addressed: The types of TB (extrapulmonary, pulmonary etc) are not presented. Some types of TB are treated for longer periods than others. Depending on when this analysis was conducted, for patients who initiated TB treatment in late 2020 and were to be treated for longer than 6 months may not have completed treatment. These could have been categorized as “not evaluated”. How did the authors ensure that only patients who should have completed treatment at the time of analysis were excluded? -The formulas highlighted that outcomes for all forms of TB were analysed. In Zimbabwe, TB treatment outcomes for drug-sensitive TB are evaluated 12 months from the time of treatment initiation, for example, those initiated on TB treatment in December 2020 had their outcomes evaluated at the end of 4th quarter of 2021 (i.e., first week of 2022). The analysis was conducted in 2022 and all outcomes analysed were for patients who should have completed treatment. This phenomenon is also described in the introduction section of the manuscript.

 The description of the reporting system for TB data should clearly state that only aggregate data is transmitted based on certain pre-defined indicators. The list of indicators that are routinely reported should be mentioned. -We have included in the description that ‘only aggregate data is transmitted based on certain pre-defined indicators. Indicators that are routinely reported include: treatment success, cure, treatment completed, failure, death, lost to follow up, and not evaluated rates

 Only a list of TB treatment outcome variables is described yet analyses include presumptive TB cases. The sources and variables for the presumptive analyses are not described. -We have included the list and source of variables for presumptive analysis in the description of variables.

 There is a mix up in the methods section. The section entitled “study variables” presents a list of how analyses were done. This section should be merged with the “data analysis” section and restructured to be a description rather than a list. -We have restructured the sections in the manuscript. Thank you for the guidance

 The data analysis section attempts to describe the presumptive TB cascade rather than describing how analyses were conducted. The paragraph talking of this cascade is also confusing with a statement suggesting that smear microscopy should be excluded in the presumptive TB cascade followed by a statement stating that smear microscopy was included. -Thank you for the observation. We have removed the smear microscopy bit because we realise adding it confuses readers and removing it will not affect our message. 

 Additional analyses could have been conducted to explore if some aggregate variables were associated with unfavourable outcomes. In the results and discussions, the authors suggest that residence in certain regions was associated with unfavourable outcomes yet no formal testing of such analyses were performed. -The study was a descriptive study. In the discussion authors gave their opinion based on literature that indicates the existence of a triple burden of TB/HIV and silicosis in mining communities, and nature of ASMs which are suggestive of poor prognosis in ASMs. 

-We included a recommendation for further analysis/ study to determine variables associated with unfavourable outcomes

 The results could have been presented in a sequential manner showing how many participants enter each step of the cascade and how many successfully transitioned to the next step. A flow diagram might have been very useful in illustrating this. The cascade could include steps such as presumptive TB, investigated, diagnosed, treatment initiated, treatment completion. This would assist the authors to better describe their results. Thank you for the advice

-We have illustrated the steps in a flow chart (Figure 2).

 The first paragraph of the discussion reads like a list rather than a high-level summary of key findings. -we take note and have restructured it to a more continuous flow of findings. 

 This study using routine data had profound limitations that should be discussed elaborately. Currently, the limitations section is too brief. -We have noted your and other reviewers’ input and have included more limitations. These include:

We have included challenges with data quality, lack of information about re-infection, misattribution bias due to use of routine data.

 Pulmonary bacteriologically confirmed Pulmonary clinically diagnosed Extrapulmonary TB 

Age Male Female Male Female Male Female Total 

<15 132 126 290 272 47 34 901

15-64 5291 2570 3207 1923 602 427 14020

65 + 292 145 374 213 77 54 1155

Total 5715 2841 3871 2408 726 515 16076

Age groups Male Female Total

<15 2 4 6

15-64 218 120 338

65+ 7 5 12

Total 227 129 356

---

## [Decision Letter · Decision Letter 1]

6 Jun 2023

PONE-D-22-27487R1Tuberculosis treatment outcomes in Zimbabwe, 2021: The need to strengthen patient follow upPLOS ONE

Dear Dr. Chadambuka,

Thank you for submitting your manuscript to PLOS ONE. After careful consideration, we feel that it has merit but does not fully meet PLOS ONE’s publication criteria as it currently stands. Therefore, we invite you to submit a revised version of the manuscript that addresses the points raised during the review process.

We look forward to receiving your revised manuscript.

Kind regards,

Frederick Quinn

Academic Editor

PLOS ONE

Reviewers' comments:

Reviewer's Responses to Questions

**Comments to the Author**

1. If the authors have adequately addressed your comments raised in a previous round of review and you feel that this manuscript is now acceptable for publication, you may indicate that here to bypass the “Comments to the Author” section, enter your conflict of interest statement in the “Confidential to Editor” section, and submit your "Accept" recommendation.

Reviewer #1: (No Response)

Reviewer #2: All comments have been addressed

2. Is the manuscript technically sound, and do the data support the conclusions?

Reviewer #1: Partly

Reviewer #2: Yes

3. Has the statistical analysis been performed appropriately and rigorously? 

Reviewer #1: Yes

Reviewer #2: Yes

4. Have the authors made all data underlying the findings in their manuscript fully available?

Reviewer #1: Yes

Reviewer #2: Yes

5. Is the manuscript presented in an intelligible fashion and written in standard English?

Reviewer #1: No

Reviewer #2: Yes

6. Review Comments to the Author

Reviewer #1: Thank you for the opportunity to review this revised manuscript. Most of my prior comments have been addressed. I have a few ongoing/additional suggestions for the authors, detailed below. The line numbers throughout refer to the authors' tracked-changed version of the revised manuscript.

General

- Some of the syntax and grammar in the revised version remains difficult to understand, and there are still a few typos in the revised manuscript. I would recommend another read-through to ensure legibility.

Introduction

- Line 82 – would define the acronym DHIS2 (even though defined in the abstract) (also , is this missing a “.3”?)

- Line 96 – possible to further define “the problem” (several problems have been introduced in this paragraph—would be helpful to specify which problem the authors are referring to)

Methods

- Line 101-103 – the data on Zimbabwe’s TB burden is already presented in the introduction; probably not necessary to repeat here.

- Line 113 – would clarify what the numbers after the provinces refer to. I think these are populations?

- Line 122 – would define DHIS2 at its first use (in line 82)

- Line 149 and throughout equations– would be consistent in how denominators are listed (“No. of all forms of TB cases registered” vs “Number of TB cases registered” vs “No. of TB cases registered”)

- Line 162 – It would be helpful to discuss who the key informants were, how they were contacted, and how these data were analyzed.

Results

- Line 188 – I’m still confused here as to how there were more outcomes evaluated than cases notified. If the authors’ point is that this discrepancy is an indication of lack of data quality, then I think this point should be explicitly stated.

- Figure 5-8 – the figure captions list that these outcomes are represented by province and district. However, it looks like only districts are shown in the map. It would be helpful to show outcomes by provinces, because the provincial level is extensively discussed in the text.

- Figure 5-8 – recommend changing “proportion” to “percentage”

- The authors mention key informant interviews in the methods, but results of these interviews aren’t explicitly reported in the results. In my mind, these results are potentially some of the most important for the authors to bring up, because they seem like they have the potential to provide important details about data quality problems.

Discussion

- Line 260-261 – I’m still skeptical of the claim that “The key findings of our study were that high death and lost to follow up rates were prevalent in provinces with artisanal and small-scale mining (ASM) activities”. Currently, the results section does not mention ASM activities, their distribution within provinces, or any statistical tests that show that provinces with high ASM activities were more likely to experience higher death and loss to follow-up than other provinces. I acknowledge the authors’ response that “the statement was an opinion, or hypothesis and a possible explanation for the findings”. However, in that case, and without any explicitly detailed evidence supporting this claim, I don’t think the authors should state that it is their “key finding”, as currently remains in the revised text.

- Because much of the discussion is concerned with the link between provinces with high ASM and adverse TB outcomes, it would also be helpful to include a map of the provinces of Zimbabwe, ideally colored by ASM concentration to illustrate the overlap with TB outcomes (which also ideally would be represented at the provincial level).

- It would also be helpful to discuss the results of the key informant interviews.

Limitations

- Line 314 – it would be helpful to define what “the usual calculations” are and how the authors’ calculations are different.

Reviewer #2: The authors have satisfactorily addressed reviewer comments. I still find the title confusing - Loss to follow up in the title sounds is tied to treatment outcomes which is not the case in the manuscript which has identified per-diagnosis loss to follow up as one of the notable results. Additionally, some of the figures - the maps are not very clear.

7. PLOS authors have the option to publish the peer review history of their article (what does this mean?). If published, this will include your full peer review and any attached files.

Reviewer #1: No

Reviewer #2: No

---

## [Author Response · Author response to Decision Letter 1]

26 Jul 2023

Comment number Editor’s comments 

General Some of the syntax and grammar in the revised version remains difficult to understand, and there are still a few typos in the revised manuscript. I would recommend another read-through to ensure legibility. 

Thank you for the feedback. The manuscript has been read through and revised

Introduction 

Line 82 Would define the acronym DHIS2 (even though defined in the abstract) (also, is this missing a “.3”?) 

The acronym has been defined. Thank you for identifying the typo the .3 was missing and has been included

Line 96 Possible to further define “the problem” (several problems have been introduced in this paragraph- would be helpful to specify which problem the authors are referring to) 

The problem has been further defined and the sentence now reads ‘This prompted an assessment of the TB treatment cascades and spatial distribution of treatment outcomes to inform programming and aid in crafting targeted interventions to the problems of unevaluated TB treatment outcomes).’

Methods 

Line 101 -103 The data on Zimbabwe’s TB burden is already presented in the introduction; probably not necessary to repeat here 

Thank you for the guidance. The information/data has been removed

Line 113 Would clarify what the numbers after the provinces refer to. I think these are populations? the numbers after the populations are populations. 

The sentence has been revised to read ‘Artisanal and small-scale miners are largely populated in Midlands province with a population of 19457 ASMS, followed by Mashonaland west 13267, Matabeleland South 12153, and Mashonaland central provinces’

Line 122 Would define DHIS2 at its first use (in line 82) 

DHIS2 has been defined at line 82.

Line 149 and throughout equation would be consistent in how denominators are listed (“No. of all forms of TB cases registered” vs “Number of TB cases registered” vs “No. of TB cases registered”) 

The formulas have been revised. We also note that we had written the standard denominator for the indicators versus the actual denominators we used in our calculation.

Line 162 It would be helpful to discuss who the key informants were, how they were contacted, and how these data were analysed. 

Thank you for the guidance. The paragraph has been revised and it now includes who the key informants were (TLCs, monitoring and evaluation officers, health information officers, TB focal persons), they were contacted through phone calls and face-to-face discussions. A thematic analysis of the responses was conducted.

Results 

Line 188 I’m still confused as to how there were more outcomes evaluated than cases notified. If the authors’ point is that this discrepancy is an indication of lack of data quality, then I think this point should be explicitly stated. 

It has been explicitly stated that lack of data quality was the main explanation for the data discrepancy per your guidance. Other reasons why more outcomes were evaluated have also been stated under responses from key informants.

Figure 5 - 8 The figure captions list that these outcomes are represented by province and district. however, it looks like only districts are shown in the map. It would be helpful to show outcomes by provinces, because the provincial level is extensively discussed in the text. 

New provincial maps have been constructed and attached.

Figure 5 -8 Recommend changing proportion to percentage.

Maps now indicate that they are percentages, not proportions as earlier indicated.

The authors mention key informant interviews in the methods, but results of these interviews aren’t explicitly reported in the results. In my mind, these results are potentially some of the most important for the authors to bring up, because they seem like they have potential to provide important details about data quality problems. 

Thank you for the guidance. The qualitative results of the key informant interviews have been included in the text.

Discussion 

I’m still skeptical of the claim that “The key findings of our study were that high death and lost to follow-up rates were prevalent in provinces with artisanal and small-scale mining (ASM) activities”. Currently, the results section does not mention ASM activities, their distribution within provinces, or any statistical tests that show that provinces with high ASM activities were more likely to experience higher death and loss to follow-up than other provinces. I acknowledge the author’s response that the statement was an opinion, or hypothesis and a possible explanation for the findings”. However, in that case, and without any explicitly detailed evidence supporting this claim, I don’t think the authors should state that it is their “key finding”, as currently remains in the revised text. The provincial maps that depict a link between provinces with ASM activities and the unfavourable treatment outcomes. 

The statement was removed from key finding.

Because much of the discussion with the link between provinces with high ASM and adverse TB outcomes, it would be helpful to include a map of the provinces of Zimbabwe, ideally coloured by ASM concentration to illustrate the overlap with TB outcomes (which also ideally would be represented at the provincial level). 

The initially submitted district maps have been replaced with the newly constructed provincial maps.

It would also be helpful to discuss the results of the key informant interviews 

A discussion on the results of the key informant interviews has been incorporated into the discussion section.

Limitations

Line 314 It would be helpful to define what “the usual calculations” are and how the authors’ calculations are different. 

The section has been revised and now reads ‘Challenges with data quality also affect the interpretation of the results and to cater for the discrepancies the indicators were calculated with a different denominator from the usual. In this study, the number of all forms of TB treatment outcomes analyzed was used as the denominator when calculating the proportions of TB outcomes instead of the number of all forms of TB cases registered.’

Reviewer #2 

The authors have satisfactorily addressed reviewer comments. I still find the title confusing – Loss to follow up in the title sounds is tied to treatment outcomes which is not the case in the manuscript which has identified per-diagnosis loss to follow-up as one of the notable results. Additionally, some of the figures – the maps are not very clear. -

Thank you for the feedback. The title has been revised to ‘Tuberculosis cohort analysis in Zimbabwe: The need to strengthen patient follow up throughout the TB care cascade.’

-The maps have been revised and have constructed provincial maps instead of district maps

---

## [Decision Letter · Decision Letter 2]

17 Aug 2023

PONE-D-22-27487R2Tuberculosis cohort analysis in Zimbabwe: The need to strengthen patient follow up throughout the TB care cascadePLOS ONE

Dear Dr. Chadambuka,

Thank you for submitting your manuscript to PLOS ONE. After careful consideration, we feel that it has merit but does not fully meet PLOS ONE’s publication criteria as it currently stands. Therefore, we invite you to submit a revised version of the manuscript that addresses the points raised during the review process.

Please submit your revised manuscript by Oct 01 2023 11:59PM. If you will need significantly more time to complete your revisions, please reply to this message or contact the journal office at plosone@plos.org. Please include the following items when submitting your revised manuscript:A rebuttal letter that responds to each point raised by the academic editor and reviewer(s). You should upload this letter as a separate file labeled 'Response to Reviewers'.A marked-up copy of your manuscript that highlights changes made to the original version. You should upload this as a separate file labeled 'Revised Manuscript with Track Changes'.An unmarked version of your revised paper without tracked changes. You should upload this as a separate file labeled 'Manuscript'.

We look forward to receiving your revised manuscript.

Kind regards,

Frederick Quinn

Academic Editor

PLOS ONE

Reviewers' comments:

Reviewer's Responses to Questions

**Comments to the Author**

1. If the authors have adequately addressed your comments raised in a previous round of review and you feel that this manuscript is now acceptable for publication, you may indicate that here to bypass the “Comments to the Author” section, enter your conflict of interest statement in the “Confidential to Editor” section, and submit your "Accept" recommendation.

Reviewer #2: All comments have been addressed

2. Is the manuscript technically sound, and do the data support the conclusions?

Reviewer #2: Partly

3. Has the statistical analysis been performed appropriately and rigorously? 

Reviewer #2: Yes

4. Have the authors made all data underlying the findings in their manuscript fully available?

Reviewer #2: Yes

5. Is the manuscript presented in an intelligible fashion and written in standard English?

Reviewer #2: Yes

6. Review Comments to the Author

Reviewer #2: Abstract

- The introduction does not flow well with the rest of the abstract unless you read the manuscript introduction. It jumps straight to the preliminary cohort analysis which seems to have been informed by other challenges. Furthermore, the high proportions who are not evaluated presented in this section are for particular cities. The need to perform a comprehensive cascade analysis is not clearly stated.

- The following sentence in the abstract may need further clarification: “We assessed the TB cohorts to determine the care cascade…”. Did they mean to say “to determine the performance of the care cascade”?

- LTFU is already defined as an outcome. The following statement becomes confusing “We calculated the percentage of LTFU at various stages..”. This statement probably refers to losses or drop offs from the cascade which should be different from LTFU, which is one of the outcomes. The need to clarify this is demonstrated by the following statement which refers to cascade losses before treatment initiation as LTFU which is also a TB treatment outcome. “We reviewed data for 107583 people presumed to have TB based on symptomatic screening and or x-ray and 21.4% were LTFU before the specimen was investigated.”

- The following statement may need to be rephrased to clearly convey the intended message: “Frequencies, means, and medians were calculated, and graphs for the cascade, treatment success, and undesirable outcomes were generated”

- There is a need to write out the results in a very clear but concise manner. The current results are not currently written in a clear an exhaustive manner. From the write up it is not clear how many patients initiated treatment. Furtheremore, at national level it is not clear what proportions experienced other treatment outcomes – cure, treatment completion, death, LTFU, not evaluated etc. This is further complicated by the mixing of national results with provinvial results.

“The treatment initiation rate was 99.1%. The national average drug-sensitive treatment success rate was 87.7% and Matabeleland South Province had the lowest treatment success rate of 77.3%. Overall, there were high proportions of not evaluated outcomes. High death rates were recorded in Matabeleland South (30.0%), Masvingo (27.3%), and Matabeleland North (26.1%) provinces.”

- The abstract conclusions are very broad. For instance, “treatment outcomes were not evaluated” does not appear to be accurate given that the authors have presented treatment outcomes. The is also reference to “restitute inter-border collaboration of TB services” which does not seem to refer to any particular results. What borders do they mean? International? How are the presented results related to “inter-border collaboration”.

- Overall, there is a need to rewrite this abstract to communicate the findings more succinctly.

Introduction

- “Cohort analysis can also provide information on TB causality” this statement does not appear to be accurate.

- There is a need to review this section to improve fluency. For instance, the second paragraph has two statements following each other that do not appear to be connected. “According to the Global TB Report of 2021, an estimated 12891 cases were missed in Zimbabwe during the year 2020. Zimbabwe has a high proportion of not evaluated TB treatment outcomes. While the first statement refers to undetected TB cases, the second refers to TB treatment outcomes, it is not clear if this is also from the Global TB report.

- I was struggling to understand the connection between the cohort review process in Zimbabwe, the referenced Global TB report and the preliminary review of cohort analyses in DHIS2.3 and how this review informed the analyses reported in the manuscript. Could the authors more clearly present this information.

- The information in the introduction seems to suggest that the high proportion of TB patients who are not evaluated is the main problem that this manuscript is focusing on. Why would this mandate an assessment of the entire cascade including pretreatment losses? The authors should include more information on the need to assess the entire cascade.

-

Methods:

- The formula for calculating LTFU seems to imply that the numerator included those lost to follow up before or during while the denominator included TB patients with treatment outcomes. This approach does not appear to be accurate. There is a need to differentiate pre-treatment loss to follow up from LTFU during treatment. In the results they are presented separately.

- The data analysis section does not include information on how qualitative data was analysed.

Results

- I had a small challenge following the results section. Whereas most manuscripts start with a description of study participants, this starts by presenting data quality. Additionally, I could not find in the methods a description of how this data quality assesment was done.

- The following statement in the results is more of a discussion of the results: “This discrepancy is an indication of lack of data quality.”

- A key finding in the qualitative data is that provinces had differences in how they handled “transfer ins”. Some evaluated yet soe did not. This particular difference is not clearly stated in the results.

- The description of study participants is very brief.

- Table 2 presents absolute numbers – could the authors also present row or column percentages depending on their message.

- The way the results are presented does not clearly bring out the authors intended message on TB treatment outcomes. While Figure 2 presents the presumptive TB cascade, there is not presentation of the TB treatment cascade. This is odd given that the methods elaborately described how TB treatment outcomes were evaluated and there is a discussion on mortality during TB treatment which is not included in results.

7. PLOS authors have the option to publish the peer review history of their article (what does this mean?). If published, this will include your full peer review and any attached files.

Reviewer #2: No

---

## [Author Response · Author response to Decision Letter 2]

5 Sep 2023

Abstract The introduction does not flow well with the rest of the abstract unless you read the manuscript introduction. It jumps straight to the preliminary cohort analysis which seems to have been informed by other challenges. Furthermore, the high proportions who are not evaluated presented in this section are for particular cities. The need to perform a comprehensive cascade analysis is not clearly stated. 

-Thank you for the guidance, an introductory statement was included ‘Globally people with tuberculosis continue to be missed each year. They are either not diagnosed or not reported which indicates possible leakages in the TB care cascade. Zimbabwe is not spared with over 12000 missed cases in 2020’.

-The need for a comprehensive cascade analysis was because the different challenges were noted in various geographical areas and we needed to determine if the problems were prevalent across the board.

The following sentence in the abstract may need further clarification: “We assessed the TB cohorts to determine the care cascade…”. Did they mean to say “to determine the performance of the care cascade”? 

Thank you for the observation. The abstract needed thorough revision. We meant to determine the performance of the care cascade.

 LTFU is already defined as an outcome. The following statement becomes confusing “We calculated the percentage of LTFU at various stages.”. This statement probably refers to losses or drop offs from the cascade which should be different from LTFU, which is one of the outcomes. The need to clarify this is demonstrated by the following statement which refers to cascade losses before treatment initiation as LTFU which is also a TB treatment outcome. “We reviewed data for 107583 people presumed to have TB based on symptomatic screening and or x-ray and 21.4% were LTFU before the specimen was investigated.” 

The whole abstract was revised and the terms clearly illustrated. There was pre-diagnosis, pre-treatment, and the treatment outcome lost to follow-up 

The following statement may need to be rephrased to clearly convey the intended message: “Frequencies, means, and medians were calculated, and graphs for the cascade, treatment success, and undesirable outcomes were generated” 

The statement was rephrased and now reads ‘Univariate analysis of the data was conducted where frequencies were calculated, and data was presented in graphs for the cascade, treatment success, and undesirable outcomes’ while tables were created for description of study participants and data quality.

There is a need to write out the results in a very clear but concise manner. The current results are not currently written in a clear an exhaustive manner. From the write up it is not clear how many patients-initiated treatment. Furthermore, at national level it is not clear what proportions experienced other treatment outcomes – cure, treatment completion, death, LTFU, not evaluated etc. This is further complicated by the mixing of national results with provincial results.

“The treatment initiation rate was 99.1%. The national average drug-sensitive treatment success rate was 87.7% and Matabeleland South Province had the lowest treatment success rate of 77.3%. Overall, there were high proportions of not evaluated outcomes. High death rates were recorded in Matabeleland South (30.0%), Masvingo (27.3%), and Matabeleland North (26.1%) provinces.” 

The results section was revised and now reads “An analysis of national data found 107583 people were presumed to have TB based on symptomatic screening and or x-ray and 21.4% were LTFU before the specimen was investigated. Of the 84534 that got tested, 10.0% did not receive their results. The treatment initiation rate was 99.1%. The national average drug-sensitive treatment success rate was 87.7% and analysis of treatment outcomes done at the provincial level showed that Matabeleland South Province had the lowest treatment success rate of 77.3% and high death rates were recorded in Matabeleland South (30.0%), Masvingo (27.3%), and Matabeleland North (26.1%) provinces. Overall, there were high proportions of not evaluated treatment outcomes.”

-Treatment outcomes were not presented at the national level because the authors felt it would be a duplication of information presented at the provincial level.

The abstract conclusions are very broad. For instance, “treatment outcomes were not evaluated” does not appear to be accurate given that the authors have presented treatment outcomes. The is also reference to “restitute inter-border collaboration of TB services” which does not seem to refer to any particular results. What borders do they mean? International? How are the presented results related to “inter-border collaboration”. 

Thank you for the observation. 

 -The statement has been revised and now reads ‘Unevaluated treatment outcomes were also prevalent and data quality remains a challenge within the national TB control program’. 

- The inter-border collaboration was removed from this section because it is no longer applicable. It was more relevant when the maps were initially presented at district level.

Overall, there is a need to rewrite this abstract to communicate the findings more succinctly.

Thank you for the guidance the abstract has been revised.

Introduction 

“Cohort analysis can also provide information on TB causality” this statement does not appear to be accurate. 

Thank you for the observation. The statement has been revised to “Cohort analysis can provide information on contributing factors to either successful or unfavourable treatment outcomes”

There is a need to review this section to improve fluency. For instance, the second paragraph has two statements following each other that do not appear to be connected. “According to the Global TB Report of 2021, an estimated 12891 cases were missed in Zimbabwe during the year 2020. Zimbabwe has a high proportion of not evaluated TB treatment outcomes. While the first statement refers to undetected TB cases, the second refers to TB treatment outcomes, it is not clear if this is also from the Global TB report. 

\\

Thank you for the guidance, the paragraph has been revised.

I was struggling to understand the connection between the cohort review process in Zimbabwe, the referenced Global TB report and the preliminary review of cohort analyses in DHIS2.3 and how this review informed the analyses reported in the manuscript. Could the authors more clearly present this information. 

The global TB report indicated missed cases in Zimbabwe, yet the preliminary review of the data indicated possible leakages in the cascade. These necessitated an analysis of the cohorts to determine the extent of leakages. The section has been revised in the main manuscript.

The information in the introduction seems to suggest that the high proportion of TB patients who are not evaluated is the main problem that this manuscript is focusing on. Why would this mandate an assessment of the entire cascade including pretreatment losses? The authors should include more information on the need to assess the entire cascade. 

Thank you challenges in other areas of the cascade have been included. 17% of presumed patients in Chegutu district did not have specimens sent to the laboratory for investigation and 11% of those who submitted specimens did not receive results. And 14% deaths in Gweru district.

Methods 

The formula for calculating LTFU seems to imply that the numerator included those lost to follow up before or during while the denominator included TB patients with treatment outcomes. This approach does not appear to be accurate. There is a need to differentiate pre-treatment loss to follow up from LTFU during treatment. In the results they are presented separately. 

Thank you for the observation. Our calculations included only those who were lost to follow-up while on treatment. The formula has been revised and now excludes those lost pre-treatment. Operational definitions were also included.

The data analysis section does not include information on how qualitative data was analysed. The inductive approach to thematic analysis was used. The raw data was coded and then classified into three themes. These themes were then defined as data quality issues, outcomes not evaluated, and incorrect reporting. 

Results 

I had a small challenge following the results section. Whereas most manuscripts start with a description of study participants, this starts by presenting data quality. Additionally, I could not find in the methods a description of how this data quality assessment was done. 

The sections have been re-arranged and the description of study participants was presented first. Data was exported from DHIS2.3 and data quality assessment was done first by profiling the data followed by cleaning where the inconsistencies were followed up and verified with respective provinces. Data was matched with hard copy quarterly reporting forms. 

The following statement in the results is more of a discussion of the results: “This discrepancy is an indication of lack of data quality.” 

Thank you for the guidance the statement has been removed.

A key finding in the qualitative data is that provinces had differences in how they handled “transfer ins”. Some evaluated yet some did not. This particular difference is not clearly stated in the results. 

Those differences were unfortunately not quantified thus they were not reported.

The description of study participants is very brief. 

Thank you for the guidance, the description has been beefed up.

Table 2 presents absolute numbers – could the authors also present row or column percentages depending on their message. 

The percentages have been included in the columns

The way the results are presented does not clearly bring out the authors intended message on TB treatment outcomes. While Figure 2 presents the presumptive TB cascade, there is not presentation of the TB treatment cascade. This is odd given that the methods elaborately described how TB treatment outcomes were evaluated and there is a discussion on mortality during TB treatment which is not included in results. 

Thank you. The study was a data set analysis. The data set does not contain details of patient care. It only contains data for the presumptive cascade, and treatment outcomes hence our leaving out the treatment cascade. The mortality is reported as an outcome and was thus reported and discussed.

---

## [Decision Letter · Decision Letter 3]

23 Oct 2023

Tuberculosis cohort analysis in Zimbabwe: The need to strengthen patient follow up throughout the TB care cascade.

PONE-D-22-27487R3

Dear Dr. Chadambuka,

We’re pleased to inform you that your manuscript has been judged scientifically suitable for publication and will be formally accepted for publication once it meets all outstanding technical requirements.

Kind regards,

Frederick Quinn

Academic Editor

PLOS ONE

Additional Editor Comments (optional):

Reviewers' comments:

Reviewer's Responses to Questions

**Comments to the Author**

1. If the authors have adequately addressed your comments raised in a previous round of review and you feel that this manuscript is now acceptable for publication, you may indicate that here to bypass the “Comments to the Author” section, enter your conflict of interest statement in the “Confidential to Editor” section, and submit your "Accept" recommendation.

Reviewer #2: All comments have been addressed

2. Is the manuscript technically sound, and do the data support the conclusions?

Reviewer #2: Yes

3. Has the statistical analysis been performed appropriately and rigorously? 

Reviewer #2: Yes

4. Have the authors made all data underlying the findings in their manuscript fully available?

Reviewer #2: Yes

5. Is the manuscript presented in an intelligible fashion and written in standard English?

Reviewer #2: Yes

6. Review Comments to the Author

Reviewer #2: The authors have made a good attempt at addressing my previous comments. The article is now in good shape for publication

7. PLOS authors have the option to publish the peer review history of their article (what does this mean?). If published, this will include your full peer review and any attached files.

Reviewer #2: No

---

## [Editor Report · Acceptance letter]

31 Oct 2023

PONE-D-22-27487R3 

Tuberculosis cohort analysis in Zimbabwe: The need to strengthen patient follow-up throughout the tuberculosis care cascade 

Dear Dr. Chadambuka:

I'm pleased to inform you that your manuscript has been deemed suitable for publication in PLOS ONE. Congratulations! Your manuscript is now with our production department. 

Kind regards, 

on behalf of

Dr. Frederick Quinn 

Academic Editor

PLOS ONE